# A Highly Sensitive Piezoresistive Pressure Sensor Based on Graphene Oxide/Polypyrrole@Polyurethane Sponge

**DOI:** 10.3390/s20041219

**Published:** 2020-02-23

**Authors:** Bing Lv, Xingtong Chen, Chunguo Liu

**Affiliations:** Roll-Forging Research Institute, College of Materials Science and Engineering, Jilin University, Changchun 130025, China; lvbing17@mails.jlu.edu.cn (B.L.); chenxt16@mails.jlu.edu.cn (X.C.)

**Keywords:** pressure-sensitive, layer-by-layer assembly, multilayer structures, piezoresistive sensors

## Abstract

In this work, polyurethane sponge is employed as the structural substrate of the sensor. Graphene oxide (GO) and polypyrrole (PPy) are alternately coated on the sponge fiber skeleton by charge layer-by-layer assembly (LBL) to form a multilayer composite conductive layer to prepare the piezoresistive sensors. The 2D GO sheet is helpful for the formation of the GO layers, and separating the PPy layer. The prepared GO/PPy@PU (polyurethane) conductive sponges still had high compressibility. The unique fragmental microstructure and synergistic effect made the sensor reach a high sensitivity of 0.79 kPa^−1^. The sensor could detect as low as 75 Pa, exhibited response time less than 70 ms and reproducibility over 10,000 cycles, and could be used for different types of motion detection. This work opens up new opportunities for high-performance piezoresistive sensors and other electronic devices for GO/PPy composites.

## 1. Introduction

Recent years, portable and flexible wearable devices have developed rapidly [1], and have been applied in motion detection [2,3,4], human-machine interfaces [5,6,7,8] and flexible robots [9], which puts forward higher requirements on flexible pressure sensors. Piezoresistive pressure sensors can convert pressure changes into resistance changes. Due to their simple structure and high flexibility, more and more attention has been paid to them.

Conductive polymers are often used for 2D pressure sensors [10,11,12]. For example, Pang and colleagues prepared strain sensors based on Pt-coated polymer nanofiber arrays. Due to the unique interlocking microstructure conduction principle, the sensors’ minimum detectable pressure was as small as about 5 Pa, but the strain range was only ≤5% [13]. White and colleagues created microchannels directly in Sylgard 184 using a laser. An alloy of gallium and indium was filled into microchannels as sensing element. The strain sensor they prepared could measure both strain and curvature, but the manufacturing of sensors was a complex process with many steps resulting a low overall device yield rate [14]. Lipomi et al. prepared transparent, conducting thin films for use as pressure and strain sensors. Single-walled carbon nanotubes were deposited into a film and cured with Ecoflex silicone elastomer. The sensor could accommodate tensile strain up to 150%, but could not measure strain in a direction perpendicular to the plane of the film. [15]. It can be seen that the 2D sensor cannot meet the sensing requirements under large deformation and high pressure. At present, rough surface microstructure design has been proven to be an effective method to improve sensor sensitivity, such as protruding microstructures [8], spherical microstructures [16], micropores [13,17], bionic microstructures [18] and so on. Studies have shown that high sensitivity pressure sensors could be prepared by simplifying the design of the microstructure to reduce the difficulty of preparation. For example, using PDMS as a template substrate, the microstructure was designed by conventional sandpaper [19,20,21], leaves [22,23,24], silk [25] and so on.

Recently, due to the higher demand for compressibility, pressure sensors based on 3D structural materials have been gradually developed. Sponge is a potential substrate for piezoresistive sensors. Due to its unique porous 3D structure, it can basically recover after deformation. The 3D structural pressure sensors can meet the sensing needs under large compression deformation. For example, Lu and colleagues used layer-by-layer (LBL) assembly to coat carbon black (CB) on polyurethane (PU) sponge to prepare a CB@PU conductive sponge, which exhibited a fast response time of <20 ms and good reproducibility over 50,000 cycles, but a low sensitivity (0.068 kPa^−1^) [26]. Inspired by the structure and functions of the human fingertip, Park and colleagues built an interlocking structure in ferroelectric films through polydimethylsiloxane (PDMS) molds, and the prepared sensors could simultaneously detect thermal and mechanical stimuli. [18].

In this work, we demonstrate a simple method for manufacturing a piezoresistive sensor with a high sensitivity based on GO/PPy@PU conductive sponge. The graphene oxide (GO) and polypyrrole (PPy) are cyclically coated on the PU sponges with layer-by-layer assembly to form GO/PPy composite conductive layers. The prepared GO/PPy@PU sensors could monitor a wide range of deformations from 2% to 75% strain (corresponding 75 Pa–15 kPa). Moreover, the GO/PPy@PU sponge sensors exhibited outstanding flexibility, a high sensitivity (0.79 kPa^−1^), fast response time (<70 ms), and good reproducibility over 10,000 cycles, endowing this material with wide potential applications in pressure sensitive devices.

## 2. Material and Methods

### 2.1. Material

The graphene oxide solution was purchased from Nanjing XF Nano CO., Ltd. The polyurethane sponge was supplied by a commercial cleaning sponge (Xijie Company). Pyrrole monomer (C_4_H_5_N) and ferric chloride crystal (FeCl_3_·6H_2_O) was provided by Sinopharm Chemical Reagent CO., Ltd.

### 2.2. Preparation of GO/PPy@PU Conductive Sponges

The PU sponges (10 mm × 10 mm × 8 mm) were washed in deionized water and ethanol for 10 min. Then, the PU sponges were immersed in a hydrochloric acid solution to attach a positive charge, which provided electrostatic attraction to the negatively charged GO sheet. The sponges were soaked in 2 mg·mL^−1^ GO solution for 10 min to complete the first layer assembly by attaching the GO layers. Then, the sponges were immersed in a pyrrole (Py) ethanol solution (0.35 mol·L^−1^) for 10 min to complete the combination of Py and GO. Finally, the sponges were immersed in an aqueous solution of ferric chloride (FeCl_3_) (0.5 mol·L^−1^) to initiate the polymerization of Py until the sponges were completely attached by the black substance. Finally, the obtained GO/PPy@PU conductive sponges were dried at 80 ℃ for 6 h to remove residual water. These steps were repeated to obtain GO/PPy composite conductive layers with different thicknesses.

### 2.3. Character

The resistance of conductive sponges was measured with 124 oscilloscope (Fluke, Washington, DC, United States). The output electrical signal of conductive sponges was recorded by a digital oscilloscope (UTD2102CEX). Microstructure was observed by JSM-5900LV microscope. The thickness and other volume parameters of conductive sponges were measured by vernier caliper and thickness gauge. The mechanical properties of the sponge sensors were tested with a multi-functional tester (AGS-X, Shimadzu; 5 N load cell).

## 3. Results and Discussion

### 3.1. Material Design

As an important derivative of graphene, flake-shaped graphene oxide (GO) possesses many carboxyl groups and hydroxyl groups. These large numbers of reactive functional groups and microstructures make GO have the potential to serve as a support matrix for composite materials. The oxygen-containing functional groups between the GO sheets have a strong polarity, which is also a reason why GO is easily soluble in water, contributing to uniform dispersion of GO in the material. In addition, the common conductive polymer polypyrrole (PPy) has high electrical conductivity, and as a filler of the composite material, natural agglomeration of GO can be avoided. GO nanosheets has a strong π-π interactions with pyrrole particles, so it is expected that a stable and uniform composite material can be obtained with the composite of GO and PPy.

GO nanosheets are negatively charged, while PPy is positively charged, so it is possible to improve the interfacial adhesion of GO and PPy on sponge fibers by electrostatic action. The GO/PPy nanocomposite conductive layers were prepared by alternately infiltrating the sponge using the charge LBL assembly process. The regulation of sensor performance can be achieved by adjusting the number of dipping coatings and the concentration of the solution.

### 3.2. GO/PPy@PU Conductive Sponge Manufacturing Process

GO can be manufactured at large scales, and it is a potential precursor for the diversified composite materials, because of its good solubility and large number of active functional groups. The formation of uniform and controllable composite conductive layer by a water-based LBL assembly is fully achievable. In this study, we achieved the complexation of GO and PPy and rapid and stable deposition on sponge fibers.

Figure 1a shows the schematic fabrication procedure of GO/PPy@PU conductive sponges. Cleaned PU sponges (Figure 1b) were first immersed in hydrochloric acid solution such that positive charge was attached to the surface of the fiber. Then, the sponges were immersed in GO solution (2 mg·L^−1^)to deposit a layer of anionic GO. Scanning electron microscope (SEM) images (Figure 1d) show that the PU sponge has a 3D network structure. The surface of fiber skeleton is very smooth (Figure 1e). After coating GO layers on the PU sponges, a thin layer with slight wrinkles was attached to the surface of the PU skeleton, as shown in Figure 1f. It could be proved that GO had been successfully deposited on the surface of the fiber. The GO layers showed wrinkled structures, because of the electrostatic repulsive interaction between GO sheets with the same charge. Subsequently, the sponges were immersed in pyrrole (Py) ethanol solution (0.35 mol·L^−1^) for 10 min. The Py monomers were adsorbed on the graphene oxide film under electrostatic attraction, as shown in Figure 1g, which indicated that the pyrrole monomers had been attached to the GO layers. Finally, the sponges were immersed in FeCl_3_ solution (0.5 mol·L^−1^) for 30 min to initiate the polymerization of pyrrole. As shown in Figure 1h, the fragment structure was completely deposited on the sponge fibers with two dipping coatings, and it was confirmed that the GO/PPy composite nanomaterial was formed. The volume of the sponge square increased macroscopically, as shown in Figure 1c, due to the coarsening of the sponge fibers. The above results imply that the GO/PPy composite conductive layers had been deposited on PU sponges by the LBL method.

Figure 2a shows the synthesis process of GO/PPy. A GO sheet with a large number of active functional groups could act as an active agent to attract Py monomers to the GO layers. With Py monomer as the core, PPy grew preferentially on GO sheets under the action of oxidants. With the GO sheet as the base and PPy as the filler, negatively charged GO and negatively charged PPy combined in space to form a fragmental structure. During the compounding process of GO/PPy, the GO layers were broken and wrapped in the PPy matrix, forming a rough fragmental surface morphology, making GO/PPy a composite conductive material with high surface area and excellent electrical conductivity. Figure 2b,c show the microstructure of the sponge skeleton with one and two dipping coatings, respectively. With one dipping coating, the composite conductive layers were not completely deposited on part of the skeleton, and the resistance was unstable during the load-unload cycle. Figure 2d shows the initial resistance of the sponge sensors with different dipping times. As the durations of dipping coatings increased, the initial resistance of the sensors decreased exponentially, which was consistent with the prediction of the percolation theory. According to the threshold effect, piezoresistive materials with a conductivity near the threshold have a higher sensitivity. Excluding the sensor with one dipping coating, the sensor with a two dipping coating was closest to the threshold, so it had the highest sensitivity. Meanwhile, the larger the initial resistance, the larger the measurement range of the sensor.

It is notable that the manufacture of GO/PPy@PU conductive sponge is very simple, the sponge is basically saturated after five cycles. Thermal reduction is not required. The sensor requires only two cycles of dipping coating to achieve maximum sensitivity, as discussed below. The simple and secure manufacturing method of GO/PPy@PU conductive sponges make it possible on large scale.

The GO/PPy@PU conductive sponge was measured for the maximum compression. Figure 3a,b show images of the original sponge and the sponge with maximum compression deformation. The obtained GO/PPy@PU conductive sponge exhibited excellent flexibility with a maximum compression of 85.5%. This was attributed to the special 3D structure of sponge. Figure 3c presents the compressive stress–strain curves of GO/PPy@PU conductive sponge under 0–70% strain. It could be seen, in the region where strain was larger than 40%, the pressure required to increase the strain of samples increased gradually because of the densification of GO/PPy@PU conductive sponge. Equation (1) was selected to fit the stress-strain relationship of polyurethane sponges during compression [27].
(1)$$\sigma =C_{1}{\left[\frac{\varepsilon }{C_{3}-\varepsilon }\right]}^{C_{2}}$$

The parameters related to material properties are as follows: $C_{1}=10.06$, $C_{2}=1.5$, $C_{3}=129.47$. The solid line in Figure 3c was the stress-strain curve of the polyurethane sponge predicted by the model. It could be seen that the experimental results agreed well with the theoretical predictions, indicating the correctness of the sponge compression model. Figure 3d illustrates the cyclic stress–strain curves of GO/PPy@PU. Strain rate was 0.2 s^−1^(2 mm/s). Data sampling frequency of the tester was 1000 Hz. Wave pattern of loading/unloading cycle was triangular. The sponge sensor had a pressure hysteresis during unloading. The larger the strain, the more obvious the hysteresis.

PU sponge has excellent compressibility and ability to recover from deformation. As a sensor structure substrate, it helps to test large strains.

### 3.3. Pressure-Sensing Mechanism

It has been proved that rough conductive microstructures could improve the sensitivity of pressure sensors. Inspired by this, we did not choose graphene as the conductive filler, but choose the fragmental GO/PPy composite. The original PU sponge had a 3D porous network structure, which was integrally connected by a randomly distributed fiber skeleton. The deposition of GO/PPy composite material on the sponge fiber by LBL avoided a wide range of intertwining or stacking, so that the conductive filler could be uniformly wrapped on the sponge fiber, which ensured the cycle stability of the prepared sensor. As shown in Figure 4a,b, the GO/PPy composite was completely coated onto the sponge skeleton. When the GO/PPy@PU conductive sponge was compressed, the bending of the skeleton caused tension on the GO/PPy conductive layers. Therefore, permanent mechanical microcracks were easily generated on the GO/PPy layers. However, the conductive layers composed of composite material were multi-layered structure, there were certain buffer gaps between the GO/PPy sheets. Therefore, when the crack was generated and enlarged, there were not many completely broken cracks generated to cause an inverse increase in resistance.

Moreover, the compression of the sponge under small deformation would lead to the rapid change of the contact between the conductive skeleton of GO/PPy and the contact area, which would cause the breakage-recovery of local conductive path, as shown in Figure 4e. Under this synergistic effect, the sensitivity of the sensor under small strain was improved.

With the further increase of compressive strain, the gap in the sponge was further reduced, and the contact quantity of GO/PPy conductive skeleton would increase to saturation. So the decisive effect on the total resistance was the contact area between GO/PPy conductive skeleton, as shown in Figure 4c,d.

According to the theory of percolation, the relationship between the resistivity of the composite material and the content of the conductive filler can be expressed by Equation (2).
(2)$$\sigma_{m}=\sigma {\left(\varphi -\varphi_{c}\right)}^{-t}$$
where $\sigma_{m}$ and σ stand for the resistivity of the conductive sponge and GO/PPy composite, φ is the content of the GO/PPy in conductive sponge, $\varphi_{c}$ represents the percolation threshold, t reveals the critical factor.

The volume resistance and volume parameters of the conductive sponge can be described by the Equation (3):(3)$$R=\sigma_{m}\frac{L}{S}$$
where R represents resistance of the conductive sponge, L and S reveals thickness and cross-sectional area of conductive sponge. Then we have the Equation (4) to describe the relative resistance change of the conductive sponge.
(4)$$\frac{\Delta R}{R_{0}}=\frac{R_{0}-R_{P}}{R_{0}}=1-\frac{L_{P}\sigma {\left(\varphi_{P}-\varphi_{c}\right)}^{-t}/S}{L_{0}\sigma {\left(\varphi_{0}-\varphi_{c}\right)}^{-t}/S}\;=1-\frac{L_{P}}{L_{0}}{\left(\frac{\varphi_{0}-\varphi_{c}}{\varphi_{P}-\varphi_{c}}\right)}^{t}$$
where $R_{0}$, $R_{P}$ respectively represents the resistance of the conductive sponge in the initial state and an arbitrary compressed state, $L_{0}$
$L_{P}$ respectively represents the thickness of the conductive sponge in the initial state and an arbitrary compressed state, $\varphi_{0}$, $\varphi_{p}$ represent the volume fraction of the conductive filler GO / PPy in the initial state and an arbitrary compressed state, respectively.

When the conductive sponge was subjected to normal force, the compression strain produced was ε.
(5)$$\frac{L_{P}}{L_{0}}=\frac{L_{0}\left(1-\varepsilon \right)}{L_{0}}=1-\varepsilon$$
(6)$$\varphi_{P}=\frac{\varphi_{0}}{1-\varepsilon }$$

Set $m=\varphi_{c}/\varphi_{0}$ substituting Equation (5) and Equation (6) into Equation (4), we have
(7)$$\frac{\Delta R}{R_{0}}=1-\left(1-\varepsilon \right){\left(\frac{1-m}{\frac{1}{1-\varepsilon }-m}\right)}^{t}$$

The conductive fibers in the conductive sponge were connected to form the 3D network, so the volume fraction of the conductive filler in the initial state $\varphi_{0}$ exceeded critical volume fraction $\varphi_{c}$, m < 1. Equation (7) describes the relationship between the relative resistance change and strain. The opposite sides of the sensor sponge were coated with conductive silver paste with copper sheets to eliminate contact resistance and obtain a stable signal output, as shown in Figure 5a. When m = 0.05, t = 2, the theoretical value showed a good fit with the experimental value, as shown in Figure 5b. It could be seen that the relative resistance change was divided into two stages. When the strain was less than 40%, the relative resistance change increased rapidly with strain, indicating that the resistance decreased rapidly. This could be attributed to the simultaneous increase in the number of contacts and the contact area of the GO/PPy conductive skeleton in this stage. However, as the applied strain gradually increased (40–80%), the relative resistance change slowly increased. Because the number of contacts between the conductive skeletons of GO/PPy was gradually saturated, the contact area between the conductive skeletons became the dominant factor for the resistance change (Figure 4e). A gauge factor (GF) was calculated, which was defined as the ratio of relative resistance change (ΔR/R_0_) to strain to evaluate the sensitivity of the GO/PPy@PU conductive sponge to the applied strain. The GF of the 0–40% strain stage was 2.1. The average GF value of the 40–80% strain stage was reduced to 0.5. These data indicated that GO/PPy@PU conductive sponges had higher sensitivity to strain in low strain regions (0–40%). Figure 5c shows the performance of different samples with two dipping coatings. Samples made in different batches showed essentially the same response to pressure, reflecting a good repeatability in the manufacturing process.

Substituting Equation (1) into Equation (7), we have
(8)$$\frac{\Delta R}{R_{0}}=1-\left(1-\frac{C_{3}{\left(\sigma /C_{1}\right)}^{1/C_{2}}}{1+{\left(\sigma /C_{1}\right)}^{1/C_{2}}}\right){\left(\frac{1-m}{\frac{1}{1-\frac{C_{3}{\left(\sigma /C_{1}\right)}^{1/C_{2}}}{1+{\left(\sigma /C_{1}\right)}^{1/C_{2}}}}-m}\right)}^{t}$$
which reveals the relationship between relative resistance change and pressure. Figure 6a shows the theoretical prediction curves and experimental results under different dipping times. The pressure sensitivity S was be defined as the slope of curves in Figure 6a (S = δ(ΔR/R_0_)/δP, where P denoted the applied pressure). The study found that the number of dipping coatings affected the piezoresistive properties of the sensors. As the number of dipping coatings increased, the pressure sensitivity of the sponge sensor increased first and then decreased. It was worth noting that the pressure response behavior of GO/PPy@PU conductive sponge could be divided into two stages. When the pressure was less than 2.5 kPa, ΔR/R_0_ increases rapidly with increasing strain, showing a sensitivity to pressure of 0.79 kPa^−1^. In this region, the increase of the number of contacts and the contact area of the GO/PPy conductive skeleton played a decisive role in the resistance change. Moreover, as mentioned above, due to the special structure of the sponge, GO/PPy@PU conductive sponge had less deformation resistance when the compression strain is less than 40%, which also increases the sensitivity of the GO/PPy@PU conductive sponge to pressure at small strains. However, when the pressure exceeded 2.5 kPa, the contact area between the GO/PPy conductive skeletons became the dominant factor, and at this stage, the deformation resistance of the GO/PPy@PU conductive sponge also increased significantly, the sensitivity to pressure decreased to 0.012 kPa^−1^.

It can be demonstrated from the Equation (8) that the larger m, the higher the sensitivity of the conductive sponge to pressure. With two dipping coatings, the minimum number of composite conductive layers deposited on the sponge resulted in a small volume of conductive fibers, a small $\varphi_{0}$, and a large $\varphi_{c}$, resulting in maximum m (m = 0.05). The GO/PPy composite conductive layers had a fragmental rough structure. Compared with other conductive fillers with smooth surfaces such as graphene, when the conductive fiber diameter is same, the GO/PPy composite conductive layers have a smaller volume and $\varphi_{c}$ is larger, resulting in a larger m.

Otherwise, it can be demonstrated from the Equation (8) that the larger t, the higher the sensitivity of the conductive sponge to pressure. Previous studies have proved that the larger the aspect ratio of the conductive material, the higher the t value [28]. Lower dipping times lead to lower GO/PPy composite conductive layers thickness, larger conductive fiber aspect ratio, and a higher t value. When dipping coating two layers, t = 2. When the number of dipping coatings is five, t = 1.3. In summary, the conductive sponge has the highest sensitivity to pressure with two dipping coatings. The resistance was unstable due to the uneven distribution of the conductive layers with one dipping coating. Therefore, the subsequent experiments were under two dipping coating conditions.

The GO/PPy sponge sensor with two dipping coatings was preferred. Figure 6b illustrates the hysteresis curves for the GO/PPy sponge sensor. For small pressure such as 500 Pa, the relative resistance change overlapped with that in the load-unload cycle for the sample, indicating a small hysteresis in the response. For a larger pressure such as 15 kPa, there existed a relatively large hysteresis in the response. $\gamma_{R}$ was used to represent the hysteresis of the sensor, which could be expressed by Equation (9).
(9)$$\gamma_{R}=\frac{\Delta R_{\max}}{R_{\max}}\;\times \;100\%$$
where $\Delta R_{\max}$ represents the maximum difference of the relative resistance change on the cyclic curves under the same pressure. $R_{\max}$ is the maximum relative resistance change. When the pressure was 15 kPa, the hysteresis of the sensor was the largest, $\gamma_{R}$ was 12.4%.

### 3.4. Piezoresistive Properties of GO/PPy@PU Conductive Sponges

The response behavior of the GO/PPy@PU conductive sponge to small repeated compressive strain changes was recorded, as shown in Figure 7a, and a stable and continuous current response was observed. Significantly, when strain was as low as 2%, corresponding to the pressure of 75 Pa, the current change could still be detected. These responses of the GO/PPy@PU sensor to small strains may allow us to identify small movements such as pulse, throat vibration and so on.

Under large cyclic strain, the sensors could still output the corresponding characteristic current signal (Figure 7b). The intensity and shape of these cyclic curves under high strain have different characteristics. The smaller the strain, the sharper the peaks would be. These characteristic response signals made the GO/PPy@PU conductive sponge able to detect and distinguish large-scale human motion, such as joint bending.

Figure 7c shows that the prepared GO/PPy@PU sponge sensor exhibited a fast response time of <70 ms. The cyclic stability of the GO/PPy@PU sponge sensor was also tested. As shown in Figure 7d, in the 10,000 load-unload cycle test at 45% strain, the response signal output was almost constant except for current offset due to mechanical fatigue and current source noise fluctuations. On the one hand, the excellent reproducibility of the sensor could be attributed to the compression elasticity of the GO/PPy@PU conductive sponge. On the other hand, the LBL method improved the interfacial adhesion between the GO/PPy@PU composite conductive layers and the sponge skeleton, which greatly contributed to the stability of the cycle.

### 3.5. Motion Monitoring

We evaluated the ability of the GO/PPy@PU sponge sensor to monitor small-scale human activities. First, connected the GO/PPy@PU sensor to the throat to record the current signal when the tester speaks different words (for example, say hello, goodbye) (Figure 8a). As shown in Figure 8b, nearly the same characteristic current curves were produced when the same word was repeated, indicating the excellent stability of the sensor. In addition, when different words were pronounced, the characteristic current curves changed because each word caused a different vibration of the throat muscles. Significant differences between characteristic current signals indicated excellent recognition performance of sponge sensors. When the tester swallowed saliva and coughed, the sensor could still provide corresponding characteristic current signal, as shown in Figure 8c. These characteristic current signals allowed people to distinguish vibration near the throat. Then, we fixed the GO/PPy@PU sponge sensor to the tester’s wrist to record the characteristic current curve of the human pulse (Figure 8d). It could be seen in the Figure 8e that the pulse frequency was approximately 72 times/min.

To further test the performance of the sponge sensor for large deformation motion detection, the GO/PPy@PU sponge sensor was fixed to the joint of the index finger to test the bending of the finger joint with different degrees. The characteristic current signals of the GO/PPy@PU sponge sensor in different degrees of bend-release motions were recorded, as shown in Figure 8f. It could be observed that as the degree of bending of the finger gradually increased, the peak value of the characteristic current signal also increased. During the bending release phase, the steady value of the current also increased slightly. This could be attributed to the fact that the sensor produced compression deformation with the bending of the finger, and the greater the bending amplitude, the greater the compression deformation of the sponge. On the other hand, the increase in current value during the bending release phase was due to an incomplete recovery of the compressive strain in the central region of the sponge sensor. The compressive strain generated by the sensor during the bending process of the finger was not uniform, and the compressive strain in the central region was larger than that of the edge portion. Therefore, the strain recovery in the central region was likely to be incomplete during the recovery process, resulting in a small increase in the current during the bending release phase. The results show that the sensor can monitor and distinguish some of the physiological activities of humans, demonstrating its potential in motion monitoring.

## 4. Conclusions

In summary, we constructed a uniform multi-layered GO/PPy coating on the sponge fiber skeleton by LBL method to produce a high sensitivity (0.79 kPa^−1^), wide test range (75 Pa–15 kPa) sponge sensor. The GO/PPy@PU sponge sensor had excellent flexibility (85.5%), fast response time (70 ms) and outstanding cyclic stability (over 10,000 cycles). The resistance change of the sensor was fitted according to the percolation theory, the optimal dipping times were deduced, and the pressure sensitivity mechanism of the sponge sensor was explained. Prepared high-sensitivity and high-compression sponge sensors can help detect human activities under different needs. The sponge sensor is comparable in performance to recently reported devices. The LBL method provides a simple method to prepare a uniform multilayer conductive composite layer. The prepared GO/PPy composite offers new opportunities for producing a wide range of low cost, high sensitivity sponge sensors.

## Figures and Tables

**Figure 1 sensors-20-01219-f001:**
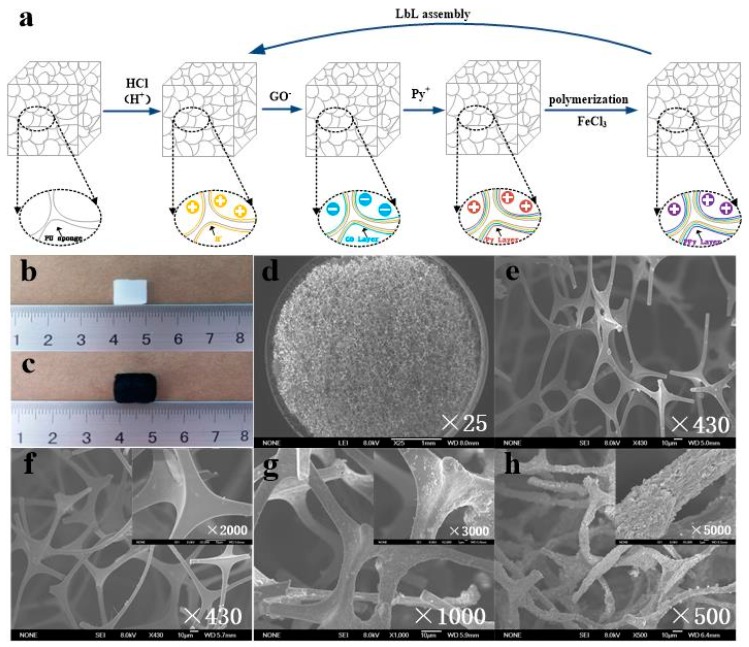
(**a**) Schematic diagram for preparation of graphene oxide/polypyrrole@polyurethane (GO/PPy@PU) conductive sponges. (**b**) Photographs of the initial PU sponge and (**c**) GO/PPy@PU conductive sponge. (**d**,**e**) Scanning electron microscope (SEM) images of the initial PU sponge. (**f**–**h**) SEM images of GO/PU sponge, GO/Py@PU sponge, GO/PPy@PU conductive sponge.

**Figure 2 sensors-20-01219-f002:**
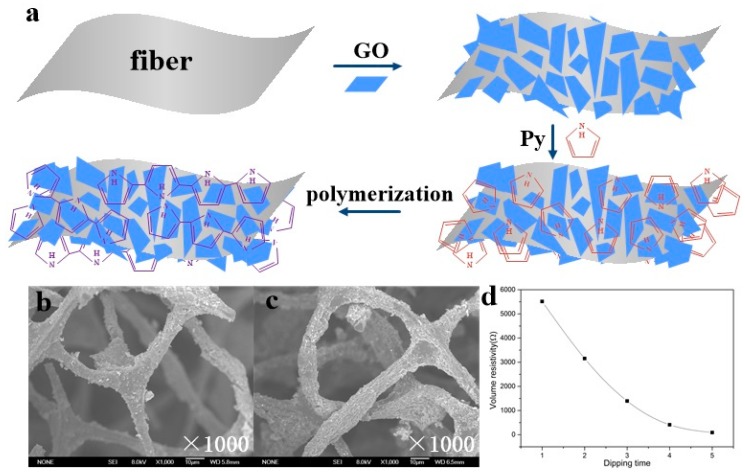
(**a**) Schematic diagram of the GO/PPy synthesis process. (**b**,**c**) SEM images of GO/PPy@PU conductive sponge with one and two dipping coatings. (**d**) Initial resistance of the sponge sensors with different dipping times.

**Figure 3 sensors-20-01219-f003:**
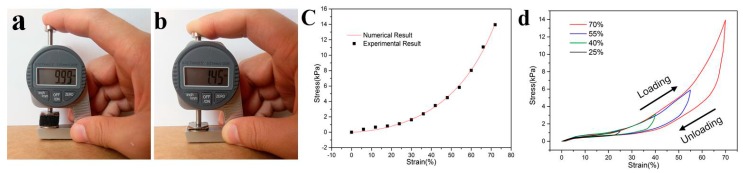
**(a**,**b**) Photographs of the maximum compressibility of GO/PPy@PU conductive sponge. (**c**) The numerical and experimental results of stress–strain curves of GO/PPy@PU with 0–75% strain. (**d**) Cyclic stress–strain curves of GO/PPy@PU.

**Figure 4 sensors-20-01219-f004:**
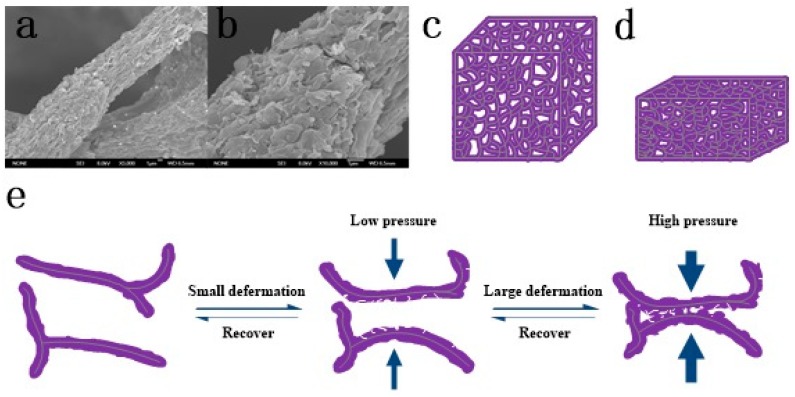
(**a**,**b**) SEM images of the microcrack caused by small strain on the conductive skeleton, magnification: (**a**) 3000× and (**b**) 10,000×. (**c**,**d**) Schematic diagram of the large deformation GO/PPy@PU conductive sponge. (**e**) Schematic diagram of GO/PPy@PU conductive sponge skeleton contact during compression.

**Figure 5 sensors-20-01219-f005:**
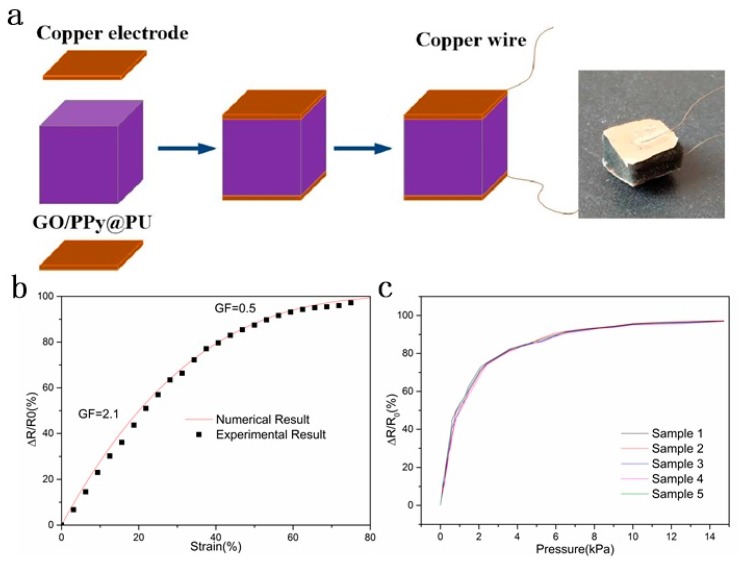
(**a**) Schematic diagram of the manufacture of sponge sensor. (**b**) The numerical and experimental results of relative resistance change (ΔR/R_0_) of the GO/PPy@PU conductive sponge and compressive strain and corresponding GF changes with two dips of coating. (**c**) Performance of different samples with two dipping coatings.

**Figure 6 sensors-20-01219-f006:**
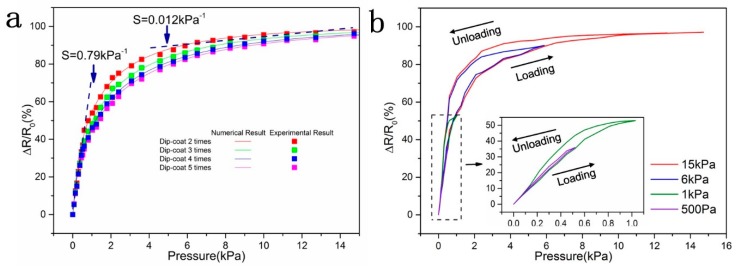
(**a**) The numerical and experimental results of the relative resistance change (ΔR/R_0_) of the GO/PPy@PU conductive sponge with two to five times of dipping the coating and gradually increasing pressure. (**b**) Hysteresis curves for the GO/PPy@PU sponge sensor.

**Figure 7 sensors-20-01219-f007:**
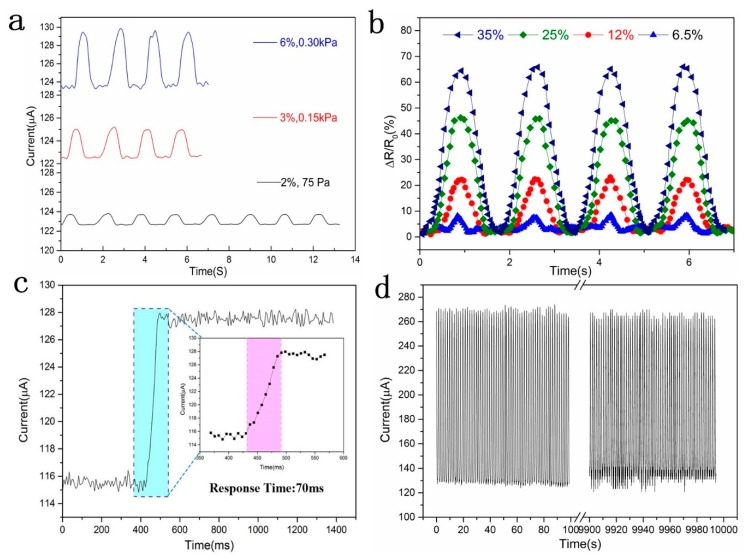
(**a**) Cyclic curves of relative resistance change of the GO/PPy@PU sponge sensor under low strain. (**b**) Cyclic curves of relative resistance change of the GO/PPy@PU sponge sensor under high strain. (**c**) The hysteresis test of the GO/PPy@PU sponge sensor with a response time <70ms. (**d**) Cyclic stability test of GO/PPy@PU sponge sensor for 10,000 cycles at 45% strain.

**Figure 8 sensors-20-01219-f008:**
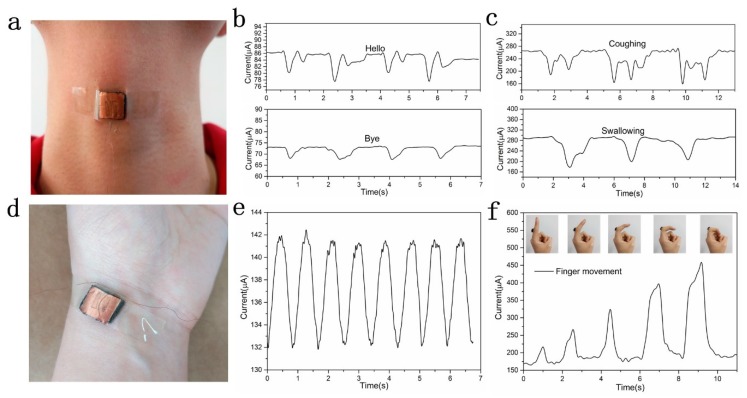
(**a**) A photograph of the GO/PPy@PU sponge sensor fixed to the tester’s throat. (**b**) Characteristic current curves during pronouncing different words, (**c**) swallowing saliva, coughing. (**d**) A photograph of the GO/PPy@PU sponge sensor fixed to the wrist. (**e**) Characteristic current curve of pulse monitoring. (**f**) Characteristic current curve of finger joint bending.

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
