# Peer review of "A Highly Sensitive Piezoresistive Pressure Sensor Based on Graphene Oxide/Polypyrrole@Polyurethane Sponge"

_sensors, 2020, doi:10.3390/s20041219_

Round 1
Reviewer 1 Report
The difference in the content of the conductive network on the sponge between using 2-5 times of dipping coating and using 1 time of dipping coating? Can it be described quantitatively, or whether the effect of this difference on the sample is considered?
The resistance of the conductive sponges? Is it possible to use percolation threshold or other theoretical quantitative analysis to validate the optimal time length and the times of dipping coating ? Ref from fig.3 in Carbon, 2018, 140, 1-9.
The effect of mechanical hysteresis on the performance of the composite?
Reviewer 2 Report
The paper is titled: "A Highly Sensitive Piezoresistive Pressure Sensors (grammatically incorrect) based on ... Sponge.
The paper is flawed and is written superficially without any scientific and international standards.
The reference list is highly skewed towards Chinese and Korean literature sources.
Equation (1) was selected to fit the stress-strain relationship of polyurethane sponges during compression[27]. --> there is no Ref 27; there are only 26 references listed.
The mathematical procedure is incorrect:
The Poisson ratio of the foam was assumed to be 0; there is no scientific proof that it is really 0
Polyurethane is a visco-elastic material; foams are energy absorbing structures; it is therefore expected that loading and unloading of the foam exhibits hysteretic behaviour; yet Figure 2c shows only a single curve; so does Figure 4b.
The mechanical properties test should be standardised with a material testing machine, instead of manual testing (Figures 2a and 2b).
Reviewer 3 Report
The paper presents an innovative wearable sensors and the application is very interesting. However, the English language should be improved for a better readibility.
Reviewer 4 Report
- Figure 1 is so complex that it is preferable to split it in two or three for the sake of understanding.
- The pressure-strain curve in Figure 2 shows nonlinearity. To be used as a practical sensor, it is necessary to linearize the calibration curve by removing nonlinearity.
- What is the evidence that GO/PPy is evenly coated on the surface of the sponge skeleton? (Quantitative data is required, not SEM photographs.)
- If the sensor developed is twisted or bent, is it possible to distort the measured value?
- Need to supplement the evidence that certain level of repeatability in the manufacturing process are guaranteed.
- It may be desirable to supplement the conclusions further.
Round 2
Reviewer 2 Report
RE: “… , and we hope the revised manuscript will meet the requirements of IONICS.”
--> your paper is reviewed by ‘Sensors’ and not by IONICS ; or did you submit this paper to both journals simultaneously?
RE: “We have studied comments carefully and have modified the manuscript according to your suggestions” -->
Your modifications seem to be superficial and written in a hurry – more details below
RE: “ The mechanical properties of the sponge sensors were tested under a multi-functional tester (Instron-5560, Boston, MA, United States). We added this description in 2.3.Charater.”
AND
RE: “The mechanical properties of the sponge sensors were tested under a multi-functional tester”
--> The description of the test conditions are missing: strain rate, data sampling frequency, wave pattern of loading/unloading cycle (triangular, trapezoidal, sinusoidal?). Which software did you use for running the Instron machine.
RE: Figures 3c,d:
--> Your maximal stress (NOT pressure!) exerted on the foam sample was 14 kPa; the cross section of the foam was 1 sqcm (Figure 1bc). The total force applied to the foam was therefore 1.4 N. Why did you use a 50 kN (50,000 N) Instron machine for testing? A 50 kN load cell is not sensitive enough for measuring maximally 1.4 N.
RE: Figures 3c,d:
--> The stress-strain curve shown in these Figures does not look like the one of a foam or cellular solid. Where is the linear elastic segment? Could it be that the force limit of the linear elastic segment was smaller than the sensitivity of the 50 kN load cell and was therefore not captured?
RE: “In this work, our purpose is to demonstrate a simple method for manufacturing a piezoresistive sensor material with high sensitivity.”
--> How was the sensitivity determined?
Via the Gauge factor? (“GF is calculated, which is defined as the ratio of relative resistance change (ΔR / R0) to strain to evaluate the sensitivity of the GO/PPy@PU conductive sponge to the applied strain.”)?
Why is then the unit of sensitivity kPa^(-1)? Strain is unitless (x/L)! Pascal is the unit of stress.
RE: “ We have modified the introduction to make the references more diverse.”
--> The only change you made in the Introduction (“all the changes are highlighted in red in the Revised Manuscript. The main corrections in the paper and”) is:
“The strain sensor prepared by Chun and 34 colleagues using single-layer graphene showed a high gauge factor (42.2), but could only work in the 35 strain range of 0-20% [14].”
which adds reference [14] to the reference list:
Reference 14 (14. Chun, S.; Choi, Y.; Park, W. All-graphene strain sensor on soft substrate. Journal of Materials Chemistry A. 2015, 3(25), 13317-13323.) was written by Korean authors.
A proper review of US American and European literature is still missing.
RE: Poisson ratio (“We reviewed the literature, modified the description and added the reference: Low-density foam materials usually exhibit negligible lateral bulging or near zero Poisson's ratio under simple compression [28].”)
--> You simply copied a full sentence from a single quoted reference; this is hardly a “literature review”:
“Low-density foam materials usually exhibit negligible lateral bulging or near zero Poisson's ratio under simple compression [28].”
What does this sentence actually prove? Is this related to your paper? Is the foam you produced a low-density one? There is no reference to the density and relative density in your paper at all.
You should refer to the Gibson and Ashby foam bible when making such statements.
Furthermore, if you quote an entire sentence by simply copying it, then you have to put it under inverted commas and preferably write it in Italic font.
As mentioned, the revision of this paper looks as if it was done in a hurry.
RE: “However, the sensor resistance changes completely with volume or strain.
Figures 5bcd are too small; it’s hard to read these figures.
RE: “The prepared GO/PPy@PU sponge sensor exhibits a fast response time of <70 ms.”
The problem of visco-elasticity (be it mechanical or electrical) is that it causes time-dependent effects with time delays and reduced response time. This means that pressure spikes are cut-off, resulting in wrong pressure data (obtained from the calibration curve of the sensor) while the response time looks deceptively perfect.
RE: “This hysteresis does not affect the response performance in the range of interest of the paper. Therefore, the effect of hysteresis on the sensor can be ignored.”
As mentioned above, the response performance might not be affected by the hysteresis, and might look deceptively perfect; yet, the calibration curve is certainly affected, which in turn delivers incorrect pressure data (magnitude of data too low through viscous damping). The ‘response performance’ is not the only electrical property of a sensor; other properties, completely ignored in this paper, are the calibration curve (pressure vs conductance) and the amount of viscosity.
From own, long time experience, conductive foams that are used as a sensor have a pronounced electrical hysteresis that cannot be ignored.
RE: “PU is a visco-elastic material; sponges are energy absorbing structures. The sponge sensors exhibited strain hysteresis behavior during unloading, as shown in Figure 3d.”
You simply copied my comments (Polyurethane is a visco-elastic material; foams are energy absorbing structures; it is therefore expected that loading and unloading of the foam exhibits hysteretic behaviour). It is very disconcerting when authors copy the reviewer’s comments (thereby claiming that these comments are the authors’ comments) without any proper literature review that supports these comments. As mentioned, the revision of this paper looks as if it was done in a hurry.
The English has not been improved yet!
Reviewer 4 Report
- Overall, the paper seems to have been revised well.
Author Response
Response to Reviewer4
Dear Reviewer:
Thank you for your useful comments on the details of our manuscript (ID sensors-675770). Those comments are all valuable and very helpful for revising and improving our manuscript.
1. Overall, the paper seems to have been revised well.
Response: Thank you again for your kind suggestions. Limited to English level, some parts may not be easy to read. We strived to make the description of the research accurate.
Round 3
Reviewer 2 Report
In order to get this manuscript to an acceptable state, 4 changes are required:
1)
RE: Strain rate was 0.2s-1 (2mm/s). Data sampling frequency of the tester was 1000Hz.
The thickness of the foam sample was approximately 10 mm (according to Figure 1bc). The maximum strain applied to the structure was 70% (according to Figure 3cd). The total deflection was therefore approximately 7 mm. With a deflection rate of 2 mm/s, it takes 3.5 s for 70% strain. As the recording frequency was 1 kHz, you should have obtained 3500 data points for each loading and unloading segment. Yet, you are showing only 1 data point every 6% of strain. This is fine for Figure 3c but insufficient for Figure 3d.
REQUIRED CHANGE:
Pls change Figure 3d from discrete to continuous, by plotting all recorded data points.
2)
RE: the unit of sensitivity is kPa^(-1)
REQUIRED CHANGE:
Pls explain your definition of sensitivity in detail and show the equation you used for calculating the sensitivity.
3)
As the Poisson ratio is not required for your model equations, there is no need for mentioning it in the first place. You did not scientifically determine the Poisson ratio and therefore it is better when you don’t mention it.
REQUIRED CHANGE:
Pls remove the following text:
The sponge sensor we prepared had a negligible Poisson's ratio at small strains. When the strain was around 50%, it showed a lateral bulge. When the strain reached 75%, the Poisson's ratio of the sponge sensor was still less than 0.1. At this time, the conductive network was also in the saturation stage, so assuming that the Poisson's ratio of the sponge sensor is zero under simple compression.
4)
REQUIRED CHANGE:
Pls change the sentence
The mechanical properties of the sponge sensors were tested under a multi-functional tester (AGS-X, Shimadzu).
to
The mechanical properties of the sponge sensors were tested with a multi-functional tester (AGS-X, Shimadzu; 5 N load cell).
